# A Systematic Review to Evaluate the Barriers to Breast Cancer Screening in Women with Disability

**DOI:** 10.3390/jcm13113283

**Published:** 2024-06-02

**Authors:** Huda I. Almohammed

**Affiliations:** Department of Radiological Sciences, College of Health and Rehabilitation Sciences, Princess Nourah bint Abdulrahman University, P.O. Box 84428, Riyadh 11671, Saudi Arabia; hialmohammad@pnu.edu.sa

**Keywords:** breast cancer screening, mammography, disability, women with disability, socio-economic factors, disparity

## Abstract

**Background:** Breast cancer (BC) is one of the leading causes of mortality worldwide. There are observed disparities in patients with disability as compared to those without disability, which leads to poor BC screening attendance, thereby worsening disease management. Aim: The aim of this systematic review is to investigate if there are disparities in screening rates in women with disability as compared to those without disability, as well as the different factors that pose barriers to patients with disability for enrolment in BC screening programs. **Method**: Using the Preferred Reporting Items for Systematic Reviews and Meta-Analyses (PRISMA) guidelines, we systematically reviewed published articles between 2008 and 2023, which assessed different factors that contributed to poor attendance in BC screening programs held across different countries. Detailed study characteristics were obtained, and methodological quality assessment was performed on the individual studies included in this review. **Result**: A total of fifty-three articles were identified as eligible studies based on the pre-defined inclusion and exclusion criteria. These included 7,252,913 patients diagnosed with BC (913,902 patients with disability/6,339,011 patients without disability). The results revealed there are demographic, clinical, financial, and service-related barriers that contributed to lower screening rates in disabled patients as compared to non-disabled. Patient age is the most common factor, with the highest effect observed for 80 years (vs. 30–44 years) [odds ratio (OR) = 13.93 (95% confidence interval (CI) = 8.27–23.47), *p* < 0.0001], followed by race/ethnicity for Hispanic (vs. non-Hispanic white) [OR = 9.5 (95%CI = 1.0–91.9), *p* < 0.05]. Additionally, patients with multiple disabilities had the highest rate of dropouts [OR = 27.4 (95%CI = 21.5–33.3)]. Other factors like education, income, marital status, and insurance coverage were essential barriers in screening programs. **Conclusions:** This study presents a holistic view of all barriers to poor BC screening attendance in disabled patients, thereby exacerbating health inequalities. A standardized approach to overcome the identified barriers and the need for a tailored guideline, especially for disability groups, is inevitable.

## 1. Introduction

Breast cancer (BC) is one of the leading causes of mortality worldwide, and it is a huge economic burden on patients [1]. The timely detection of BC can reduce mortality. Screening for BC every 2 years in women aged 50 to 74 reduces BC deaths by 26% and causes a reduction of 29% in spreading to other parts [2]. Mammography proves to be an effective way to detect BC at an early stage, thereby reducing late effects and increasing survival. Mammography is known to reduce BC mortality by approximately 20–25% [3]. However, there is evidence of existing disparities even in well-established cancer screening programs, due to which BC mortality is an unmet challenge [4].

Disability, in any form, is one of the common contributing factors for poor BC screening uptake [4]. Disability can include physical, sensory, developmental, psychological, cognitive, learning-related, and mental illness. People with disabilities face different barriers to attending cancer programs—physical barriers like accessibility to screening; financial barriers like affordability, insurance coverage, communication barriers, and unsupportive healthcare professionals; and psychological barriers like choices to opt for screening techniques, and so on [5,6]. Two other common barriers are low income and education which equally contribute to low screening attendance [7,8]. Reports from different countries highlighted this gap in cancer screening programs, e.g., women with disability in the UK are 36% less likely to participate than women without disability [9], and the prevalence of mammography uptake in women with disability in the USA was also lower [10], likewise in France, Canada, and Australia [11,12,13].

The reasons behind such disparities are large and heterogeneous across different populations and countries, but at large they are due to public health system-level inadequacies or patient –healthcare support barriers. The United States (US) government’s Centers for Disease Control and Prevention (CDC) identify physical accessibility as the primary barrier for disabled women undergoing mammography, and they provide tips and support to help overcome this obstacle [14]. System-level changes can be improvised with authoritative intervention. However, individual healthcare professionals’ support is also warranted [15]. Studies published previously with similar aims report barriers like education, employment, age, income, screening history, disability level, and geographical region as the most highlighted. A recent study concluded attitudinal, programmatic, financial policies, physical, societal, and communication approaches can be improved to encourage BC screening [16]. In order to bridge this widening gap in adherence to screening, programs should focus primarily on patients with physical disability to equalize access to screening centers, mammography equipment access [17], and reasonable support from healthcare staff. In addition, it is crucial to ensure accessibility for patients with disabilities across all socio-economic levels. This includes those from various ethnic backgrounds or rural areas who encounter barriers to accessing facilities, as well as low-income or less-educated disabled patients who may be unaware of available policies, public health benefits, and insurance coverage. The current approach often results in poorer outcomes for these groups. This justifies the urgent need for changes in public health policies and programs for better inclusivity and equal access. This is an unmet challenge and hinders the management of BC as well as increasing the treatment gap. The aim of this study is to systematically review all the available evidence to determine the burden of different risk factors that widen the gap in the uptake of BC screening services among women with disability based on different ethnic groups, ages, and types of disability. 

## 2. Material and Methodology

This is a systematic review of the published articles that assess the barriers in BC screening uptake research in women with and without disabilities. The study employed various approaches, including the Preferred Reporting Items for Systematic Reviews and Meta-Analyses (PRISMA V.5.14.0) [18] and the guidelines for Meta-Analyses and Systematic Reviews of Observational Studies (MOOSE V.3.1.1), to facilitate the conduct and dissemination of the outcomes [19].

### 2.1. Data Source and Search Strategy

A methodological systematic search in MEDLINE, Web of Science, and Scopus was performed using the medical subject heading (MeSH) terms “breast neoplasm”, “screening”, “breast cancer”, and “disable*”, with AND/OR Boolean operators to identify all the studies that discuss BC screening methods in disabled patients [18,19,20]. A population, intervention, comparison, and outcome of the study (PICOS) outline (population: patients with BC; intervention/exposure: underwent BC screening methods; comparisons: comparison between disabled and non-disabled BC patients; outcomes: the prevalence of disabled patients opting out of BC screening and the associated risk factors) was designed.

### 2.2. Study Selection Criteria

The author separately conducted the study selection process in two different steps from relevant articles published between 1 January 2008 and 7 December 2023. All articles published before that date were excluded (n = 958). All review articles, editorials, news, technical reports, book chapters, conference proceedings, and literature alike that were not original research articles were excluded (n  =  797). Additionally, duplicate articles (n = 1152) and articles published in languages apart from English (n = 155) were rejected. Initially, the articles were sorted by titles according to search results for relevancy. Second, all of the articles that were initially screened had their abstracts retrieved and evaluated based on the inclusion and exclusion criteria. The inclusion criteria for including articles for this study were as follows: the studies included disabled patients where disability was diagnosed by some clinical criteria/interview; the type of disability was defined; the study included a comparison group—disabled vs. non-disabled; the BC screening method was mentioned (mammography, breast self-examination (BSE), or clinical breast examination (CBE)); all the epidemiological studies, demographic survey reports, randomized control trials, and observational studies that had relevant data were included; and studies reporting the ethnicity/population of the reported samples. The exclusion criteria included articles that focused on training or educating BC patients about screening methods, studies estimating barriers based on caregivers’ interviews; and studies lacking quantitative data to evaluate risk factors or BMI-related disability, which are not included in this study. The cross-references of the articles that were ultimately determined were additionally looked up for other pertinent articles.

### 2.3. Data Extraction and Quality Assessment

The extraction of data was performed by the author and filtered. If the same authors have published more than once, only the most recent study or article has been used which includes independent patient cohorts. Study characteristics like the authors, year of publication, studied population, study design, disability type, disability diagnosis, screening method, age (in range) of the study cohort, sample size (with and without disability), follow-up period, risk factors, their effect size (estimated in odds ratio (OR) with 95% confidence interval (CI)), and the *p*-value were recorded. In the absence of a documented study population, the source population according to the nation of the study was taken into consideration. All univariate p-values corresponding to the effect size estimates were considered; in case of age/sex adjustment, it is explicitly mentioned.

Using criteria established by the Newcastle Ottawa scale (NOS) for case –control study quality assessment [21], the author independently evaluated the methodical efficacy of all the chosen articles. In a case–control study, determining the exposure or result of interest is the first step; in a comparison study, it is the second. With the exception of comparability, where a maximum of two stars can be provided, each parameter in the study was given one star. Therefore, the quality of the included research was assessed using a cumulative score that was assigned to each study. After much debate, a consensus was established on the scores that were in dispute. When a study received six or more stars, it was considered to be of excellent quality.

All the studies with less than 6 stars were deemed poor and eliminated from the analysis, reflecting their poor quality of the individual study in the assessed criteria (e.g., study group recruitment, assessment of outcome, etc.).

## 3. Results

### 3.1. Search and Study Selection

The process of finding and choosing studies is mapped out in Figure 1. An overall total of 7877 articles were found; 1152 of those studies were deemed to have duplicates. From the remaining articles (n = 6725), studies were excluded because they were either gray literature (n = 797) (e.g., reviews, systematic review/meta-analysis, conference proceedings, etc.), articles in other languages (n  =  155), or published before 2008 (n = 958). In total, 4652 articles were further excluded after the inclusion and exclusion criteria were applied to the 4815 remaining papers and they were screened based on title and abstract: 304 and 41 articles were book chapters and case reports, respectively, but not research articles; 644 articles did not include patients with disability; 88 studies were on in vitro models; 267 articles were on diseases other than BC; 743 were review articles; and lastly, 2570 articles did not discuss BC screening methods. Finally, on full-text screening of the remaining 158 articles, 53 were finally included in this systematic review, as 105 articles were excluded because they did not meet certain criteria. These criteria were comparing the outcomes with non-disabled patients, assessing the outcomes of educational or vocational training programs in BC patients, including interview data of family members or caregivers, and providing qualitative data. The detailed reasoning for the exclusion of 105 studies in the full-text screening steps is tabulated in Appendix A. Finally, 53 cohort studies that met the pre-defined criteria were included in the quantitative analysis.

### 3.2. Study Characteristics

The methodological and demographic characteristics of the included studies are summarized in Appendix A. A total of 7,252,913 patients diagnosed with BC (913,902 patients with disability and 63,390 patients without disability) were screened through different screening methods from which 53 studies were included in this systematic review. The age range of the included participants was mostly between 20 and 75 years, except [22], which included participants above 18 years, and included geriatric patients between 66 and 94 years, between 15 and 75 years [23,24]. Most of the studies included were cohort studies (n = 19) or cross-sectional studies (n = 16). Apart from them, eight were community-based participatory studies, six were randomized controlled trials (RCT) or part of some RCT, three were descriptive studies, and one study was exploratory. All of the included studies used mammography as the screening method, whereas three studies included other methods like biopsy, BSE, or multiple techniques. The disability status of the BC patients recruited in individual studies was diagnosed by the International Classification of Diseases (ICD)-9 or 10, International Classification of Functioning, Disability, and Health (ICF), or by a questionnaire or interview-based assessment organized by physicians, psychiatrists, or guidelines set by the trial groups or a country-specific authority organization. The disability types in the included studies were different—physical, mental, psychological, cognitive, sensory, functional, learning, or movement disability. One article each included disability due to Down’s syndrome and spinal cord injury [25,26]. The follow-up period for the recruited subjects in the studies was mostly 2 years or above. Parish SL et al. (2012) reported the lowest follow-up being 8 weeks and the highest being 4 years [27,28]. Americans constituted the major population of the studied subjects, comprising 60.37% (n = 32), followed by Europeans at 22.64% (n = 12), South Asians at 9.4% (that included East Asian subjects only), and lastly, South Americans and Australia/New Zealand residents with 3.77% (n = 2) each.

### 3.3. Methodological Quality

The cumulative quality assessment scores obtained by individual studies are represented in Appendix A. On the NOS scale for quality assessment of cohort studies, thirty-eight of fifty-three articles were deemed as good quality with a cut-off score of nine, one article was awarded eight, whereas seven and four articles were awarded seven and six, considering them of moderate quality. None of the articles were deemed to be of poor quality, and thus none were removed from the systematic analysis based on quality. 

### 3.4. Risk Factors Associated with Breast Cancer Screening in Patients with Disability

The outcome of interest in this review is the uptake of BC screening services in women with disability (Appendix A). Only those studies where the patients adhered to the program were considered. Out of all the research that was part of this evaluation, 80% (n = 42) of the studies obtained information on the use of BC screening from national or central databases or trial programs where the diagnosis was connected to the status of a disability. Survey-based measurement, or self-reporting via a questionnaire or interview, was the alternative approach. The outcome of the study was assessed from the included studies in the OR with a 95% confidence interval to estimate the measure of association between disability status and the utilization of breast screening services. Most of the studies report univariate p-values, except one study where the p-values were age adjusted [29]. Three study reported it in a ratio [30,31], When comparing women with and without disabilities to determine the usage of cancer screening services, no OR was provided because two studies failed to find any significant differences between the groups [32,33]. There were different comparison groups across studies. The most common is between disability vs. no disability, across the number of disability types as well as their severity. Other socio-economic factors like income groups, urbanization/place of residence, living conditions (living alone or with partner or family), ethnic groups, education, BMI groups, smoking, and insurance coverage were also considered for comparison. Older age is the most common factor in group participants to observe poor screening attendance. Other criteria based on which the participants were grouped were BC stages, presence of other types of chronic/catastrophic illness, intrapersonal decisions, cancer awareness, facilitator or communication, or mobility barriers.

There are several demographic, clinical, financial, intrapersonal, and service-related barriers that contribute to lower screening rates in disabled patients as compared to non-disabled. Figure 2 and Appendix A describe them in detail. Age is the most commonly studied factor, considered in 58.5% of the total studies with the largest effect known, OR = 13.93 (95%CI = 8.27–23.47), *p* < 0.0001, in a complex activity limitation (CAL) group in patients above 80 years (vs. 30–44 years), and the individual estimates ranged from 13.93 to 0.384 [34]. This was followed by race/ethnicity studies in 47.2%, disability status and education in 41.5% each, income in 37.73%, and marital status and health insurance coverage in 32.1% each. The highest odds of race/ethnicity contributing to the outcome were reported between Hispanic vs. non-Hispanic white patients, OR = 9.5 (95%CI = 1.0–91.9), *p* < 0.05 [35]. Patients with multiple disabilities had the highest rate of drop-offs from screening programs as compared to those with one disability, OR = 27.4 (95%CI = 21.5–33.3), while in those with one disability, the highest odds were in patients with Down’s syndrome (vs. no disability), OR = 11.44 (95%CI = 9.42–13.89), but the p-value was age-adjusted in this case. Other factors included education with junior college (vs. <high school), OR = 3.52 (95% CI = 0.39–31.66), and an income between net USD 38,200–45,800 (vs. <USD 15,840), OR = 2.72 (95%CI = 2.06–3.60), *p* < 0.001. The detailed effects of other risk factors contributing to the outcome of the study are represented in Appendix A. However, due to the large heterogeneity (based on comparison groups, disability type, ethnicity, diagnostic criteria, study design) across studies, we could not cumulate the effect for easier interpretation.

## 4. Discussion

There have been several studies, particularly systematic reviews and meta-analyses, that assessed the several barriers to BC screening in disabled patients in various country-specific programs [36,37]. Despite multiple efforts to include disabled BC patients in various screening programs, a significant fraction of this population generally experiences a delay in screening or remains undiagnosed for a long time. This leads to more significant challenges by decreasing the survival rate, worsening the prognosis and treatment outcomes, and increasing the financial burden. This systematic review aims to analyze the different barriers that are risk factors in such screening programs, especially for women with disability. This review clearly estimates the different demographic, clinical, financial, as well as accessibility barriers to health care services, increasing the odds of non-attendance in screening programs. Substantial heterogeneity was found across all findings, unexplained by study quality, adjustment for covariates, disability type, and diagnostic criteria, as well as policies in adherence to screening programs.

Several previous reviews have discussed such barriers in detail but did not perform a quantitative analysis of such risk factors to estimate a global scenario [38,39]. The results of this systematic analysis revealed that there was a large disparity in screening rates between patients with disability and those without disability. This gap widened with the severity of disability and type of disability, as well as the presence of other comorbidity/chronic diseases. Since there was a large heterogeneity in the included studies based on the cohort groups recruited, study design, disability diagnosis criteria, and population, we could not perform a meta-analysis of the same and ended up with a systematic review. This heterogeneity provided significant insights into the disparities in screening programs. For example, the included studies with different study designs determine the statistical data of the recruited cohort in the analysis at a specific time point of the disease course. This can subsequently incur bias in the qualitative analysis of the systematic review. In addition, articles with qualitative data were excluded from this systematic review as the reported effect of each barrier between disabled and non-disabled groups could not be cumulated. However, the insights and complications of the obstacles that women with disabilities encounter when undergoing BC screening in such studies may be useful to better understand the disparities in this study.

The literature in this study evaluating women with disability across different ethnic groups/countries indicated that there are three prominent barriers to BC screening: sociodemographic, health insurance, and physical barriers. Socio-demographics, health insurance, caregivers, facilitators or assessment parameters, and physical barriers impair access for disabled women to BC screening, which is a vital measure in the timely detection of breast-preventable morbidity and mortality. Older age contributed to most drop-out cases; women who are 65 years or older participate less in screening programs. We also found participation varied with the type and severity of disability, as well as the presence of other chronic diseases. Physical barriers like accessibility to cars, living conditions, facilitators, or health assessments were associated with a further reduction in the likelihood of participating in breast screening, as they are for women in general. One of the essential parameters was health insurance coverage and benefits received. Patients with no benefits generally dropped off the screening programs. Additionally, intrapersonal decisions like emotional support, knowledge about BC and mammography, and language/communication barriers added to the outcome of the study. Measures are needed to address these limiting factors for women with disability so that they can actively participate in screening programs rather than being marginalized by their disability status.

The included studies discuss disability to be one of the major barriers to BC screening, with the type of disability and severity of disability, as well as associated secondary diseases (and their nature—catastrophic or chronic), being equally essential barriers. Nonetheless, it was difficult to estimate which type of disability contributed the most towards poor screening records because around 30% of the included studies recruited patients with multiple disabilities including physical, sensory, cognitive, learning, and others. The other 20% of the studies were on BC patients with intellectual disability (ID), with or without other developmental disabilities. Another 10% each were on patients with sensory issues or disability related to mental disorders. And the remaining studies included patients with physical, movement, learning, and cognitive disability, multiple sclerosis, Down’s syndrome, spinal cord injury, and cerebral palsy. Though the physical barrier was one of the crucial ones in all types of disability, due to heterogeneous data, we were unable to distinguish one disability type as the significant barrier to BC screening. 

These findings further emphasize the urgent need for an inclusive cancer screening program, awareness, or training to increase participation in such programs and overcome the accessibility to avail healthcare services with assisted guidance and facilitation. There is also a need to appropriately design such studies/interventions to improve cancer screening uptake among women with disabilities across any geographical region, as well as beyond socio-economic barriers. Providing vocational training to caregivers, encouraging screening uptake, and ensuring facilities and information are accessible—these interventions should address the common hurdles encountered. Regulatory policies and governing authorities must relax the guidelines for disabled patients to ensure inclusivity. Otherwise, women with disabilities will still suffer from reduced screening rates and preventable cancer deaths until these reforms are implemented.

Despite country-specific national cancer screening programs and policies attempting to equalize efforts for the ‘right to health’ in terms of medical services, health education, treatment of disease, and a mandate for medical facilities considering the type and severity of disability, the results of the current study demonstrated the widening gap of disparities existing in BC screening programs [39,40,41,42]. This underlines the need for policies about health awareness—accessible information and education, accessibility to screening centers, equipment, healthcare professionals sensitized to manage patients with disability, and technical support, as well as rehabilitation/vocational training programs. Screening care providers should receive disability awareness training as part of these interventions; caregivers should encourage patients to promote screening; and facilities and information should be made accessible. In spite of the active measures taken to overcome the physical or regulatory barriers of authorities or government bodies at screening centers, patients were observed to face other intrapersonal barriers like a lack of confidence and communication and psychological issues like anxiety or depression, which withdrew them for participation. It is thus ideal to provide facilitators for patient support and ease of access. 

There are several limitations to this study. The level of heterogeneity in the data curated in the independent studies is large and likely because of different disability types, their diagnostic criteria, and their assessment scales in disability score and severity. The various statistical models used in different studies also added to the heterogeneity. To overcome such heterogeneity was beyond the scope of this review. Several studies have assessed disability based on interviews or guided questionnaires, which are not accurate and vary from clinician to clinician. Several studies have associated disability severity as a barrier, but they were not estimated by a unified scale. Due to varied study designs, the raw data could not be cumulated. Apart from these, there are a few strengths to this study. This is the first study to provide a holistic view considering all the factors that might be a barrier to BC screening across the globe and across different countries. We also considered multiple types of disability and discussed the barriers in each type of disability. Performing this systematic analysis following the PRISMA and MOOSE guidelines added to the methodological rigor. None of the included studies were deemed of poor quality, thereby enhancing the overall quality of this analysis. 

In conclusion, having any morbidity reduces the odds of BC screening uptake and this is particularly pronounced for individuals with chronic disease and disability conditions. This systematic review performs a rigorous analysis across different populations as well as different types of disability. It estimates demographic, clinical, financial, and service-related barriers as major contributors to lower screening rates in disabled patients as compared to non-disabled patients. Efforts to reduce such disparities for BC in women with disabilities should focus primarily on improving accessibility by improving standards to remove physical barriers like mobility and access to screening centers, equipment, and healthcare facilities. Secondly, this can be achieved by addressing healthcare professional bias—in terms of support, preventive care, and vocational training to increase awareness about cancer. And lastly, regulatory bodies should focus on overcoming socio-economic barriers to equally dispense the national policies across all social and ethnic, as well as economic, strata. Another way is to increase the representation of women with disabilities in clinical trials focused on estimating this outcome in screening practices, which may give valuable insights into reducing treatment-related disparities. Instead of relying solely on screening attendance, future studies should investigate why some groups’ members are less likely to show up, to be able to pinpoint targeted interventions, with an emphasis on measuring informed decision-making.

## Figures and Tables

**Figure 1 jcm-13-03283-f001:**
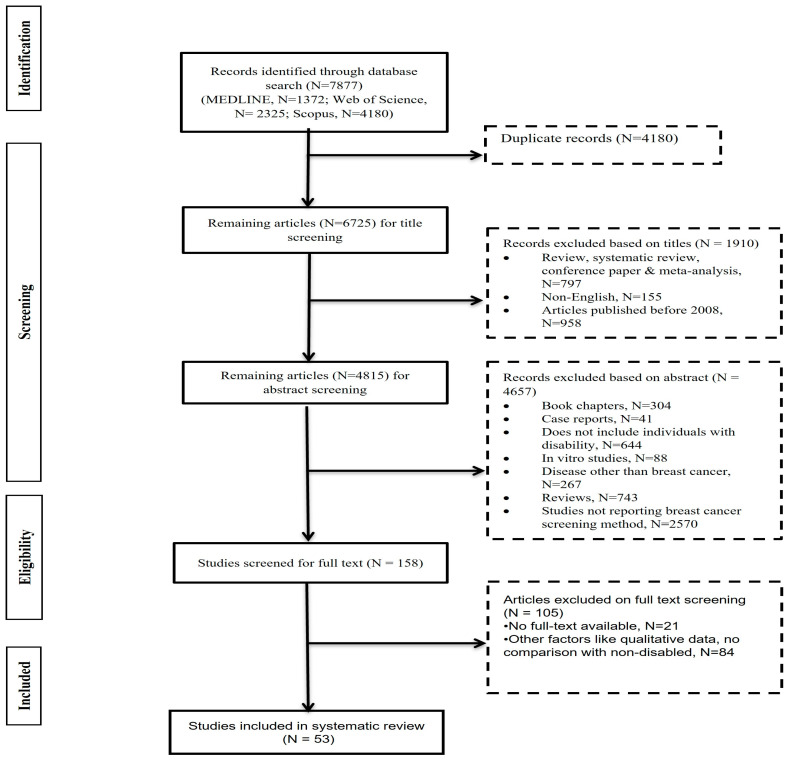
PRISMA flowchart for the performed systematic review.

**Figure 2 jcm-13-03283-f002:**
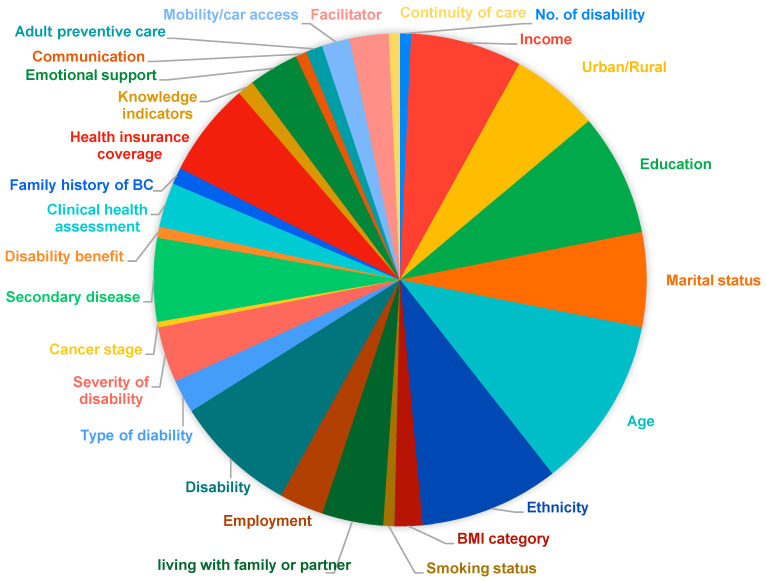
Pie chart representing the type of barriers in BC screening estimated in the included 53 studies.

## Data Availability

Not applicable.

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
