# Peer review of "A Systematic Review to Evaluate the Barriers to Breast Cancer Screening in Women with Disability"

_jcm, 2024, doi:10.3390/jcm13113283_

Round 1

Reviewer 1 Report

Comments and Suggestions for Authors

Numbers of the included patients should be corrected.

Abbreviations should be checked.

Author Response

Response to Reviewer’ Comments

All the page and line numbers for each response to reviewers’ comments corresponds to the file named “jcm-3003191-track changes.doc”

Reviewer 1:

  1. The numbers of the included patients should be corrected.

We thank the reviewer for finely scrutinising our manuscript. As per the comment, the number of included patients have been corrected. See corrections at

Section: Abstract, Page: 1, Line: 27, 28

Section Result 3.2. Study characteristics, Page: 7, Line: 193,194

  1. Abbreviations should be checked.

We appreciate the reviewer for taking such a detailed effort to review our manuscript. The abbreviations have been included in the respective places. A few examples can be seen in:

Section: Abstract, Page:1, Line:21,31

Section: Abstract, Page:2, Line:32

Reviewer 2 Report

Comments and Suggestions for Authors

The author has conducted a systematic review evaluating the barriers in breast cancer screening in women with disability, utilizing available literatures and adhering to the standard guideline (PRISMA) for systematic reviews. The study shows that the lower screening rates for breast cancer in patients with disability due to many factor including socio-economic factor, but the details on major disabilities for the poor attendance for breast cancer screening  remain elusive. 

Minor corrections: 

Write the expansion of following abbreviations in the abstract: “PRISMA” “QR” “CI”

Figure 1 requires further attention. All words appear in different fonts and size, giving the impression that they have been sourced from various places. 

Comments on the Quality of English Language

The article is well written, but needs some editing. The English language in this article is readable and understandable. 

Author Response

Response to Reviewer’ Comments

All the page and line numbers for each response to reviewers’ comments corresponds to the file named “jcm-3003191-track changes.doc”

Reviewer 2:

  1. The author has conducted a systematic review evaluating the barriers in breast cancer screening in women with disability, utilizing available literatures and adhering to the standard guideline (PRISMA) for systematic reviews. The study shows that the lower screening rates for breast cancer in patients with disability due to many factor including socio-economic factor, but the details on major disabilities for the poor attendance for breast cancer screening remain elusive.

We are obliged to the reviewer for their valuable comment. In this systematic review the included studies comprise of BC patients with different types of disability and hence we could not quantitate which type of disability is a major contributor for poor attendance in BC screening programs. We have discussed this in the following section.

Section: Discussion, Page:11-12, Line:309-320

Minor corrections:

  1. Write the expansion of following abbreviations in the abstract: “PRISMA” “QR” “CI”

We appreciate the reviewer for taking such detailed effort to review our manuscript. The abbreviations have been included in the respective places. A few examples can be seen in:

Section: Abstract, Page:1, Line:21,31,32

  1. Figure 1 requires further attention. All words appear in different fonts and size, giving the impression that they have been sources from various places.

We appreciate the reviewer for their comment. The figure has not been modified and a new figure is added. The different font and sizes seems to be because of change in the file formats. 

See file “Figure1-Revised.doc”

Reviewer 3 Report

Comments and Suggestions for Authors

The systematic review titled “A systematic review to evaluate the barriers in breast cancer screening in women with disability” by Huda I Almohammed analyzed 53 studies spanning from 2008 to 2023. The primary objective of this study was to investigate the disparities in screening rates in women with disability as compared to those without disability, as well as the different factors that pose barriers. It is a very interesting and well-written paper that provides a valuable insight that significantly enrich the existing body of knowledge in this crucial area of research.

Author Response

Thanks for your time 

there are no comments 

Dr.Almohammed

Reviewer 4 Report

Comments and Suggestions for Authors

This review is interesting. However, for the improvement need some revisions

1. Check grammatical error throughout the manuscript

2. The author mentioned that a total of 53 articles used. However, only 43 articles were cited. What about remaining articles.

3. Revise the Introduction part

4. Include some figures for better understanding of general audience

5. Discussion and conclusion parts must be improved 

Comments on the Quality of English Language

Moderate editing of English language required

Author Response

Response to Reviewer’ Comments

All the page and line numbers for each response to reviewers’ comments corresponds to the file named “jcm-3003191-track changes.doc”

Reviewer 4:

  1. Check grammatical error throughout the manuscript.

We thank the reviewer for the comment. The manuscript has been thoroughly checked for grammatical errors. 

  1. The author mentioned that a total of 53 articles used. However, only 43 articles were cited. What about remaining articles.

We appreciate the reviewer for their comment, the remaining articles included in the systematic review are now cited in the manuscript. These references are now added in Table 2 against each included study of this systematic review.

Please see reference no.45-81

  1. Revise the Introduction part

As per the reviewer’s comment, the author made a dedicated effort to improve the introduction section. A few examples of modified texts in the revised introduction can be seen at

Section: Introduction, Page:3, Line:68-70, 72-75

  1. Include some figures for better understanding of general audience

We indeed agree with the reviewer that a figure should be included for a holistic representation of this study which aims to assess barriers in BC screening in patients with disability. Therefore, we added supplementary table 4 as well as figure 2 to sum it up.

Please see figure 2

  1. Discussion and conclusion parts must be improved

The author is truly indebted to the reviewer for their kind efforts to revise this manuscript. According to the reviewer’s comment, the author made a dedicated effort to improve the discussion and conclusion section. A few examples of modified texts in the revised text can be seen at

Section: Discussion, Page: 11-12, Line: 309-320.

Section: Discussion, Page: 12, Line: 341-345.

Section: Discussion, Page: 13, Line: 361-364.

  1. Moderate editing of the English language required

We thank the reviewer for the comment. The manuscript has been thoroughly checked for English language errors and has been edited and improved. 

Round 2

Reviewer 4 Report

Comments and Suggestions for Authors

I recommend to accept for publication